# Research on rock breaking mechanism of PDC cutter under the action of ultrasonic vibration

**Ruocheng Zhang[1], Zhanfang Huang[1,2], Zengzeng Zhang [1]\*, Yalu Han[3], Zhendong Wang[4], Chunguang Wang[1], Qing Yan[1]**

1 School of Civil Engineering and Geomatics, Shandong University of Technology, Zibo, China, 2 Key Laboratory of Highway Construction & Maintenance Technology in Loess Region, Shanxi Transportation Research Institute, Taiyuan, China, 3 Department of Construction and Engineering, Shandong Cancer Hospital and Institute, Shandong First Medical University, Shandong Academy of Medical Sciences, Jinan, China, 4 Key Laboratory of Coal Exploration and Comprehensive Utilization, Ministry of Nature and Resources, Shaanxi Coal Geology Group Co., Ltd., Xi'an, China

\* zhangzengzeng@sdut.edu.cn

**Data Availability Statement:** All relevant data are within the manuscript and its Supporting Information files.

## Abstract

Ultrasonic vibration technology has significant potential for breaking hard rocks. Understanding the optimal frequency for rock breaking under ultrasonic vibration can significantly reduce the cost of rock breaking and extend the service life of polycrystalline diamond compact (PDC) cutters. This is important for practical engineering applications. This study presents a three-dimensional finite element model of rock breaking by a PDC cutter under ultrasonic vibration. The model was established using ABAQUS software and used to simulate the dynamic rock breaking process of the PDC cutter. A comparative analysis was performed between conventional rock breaking and rock breaking under ultrasonic vibration. According to the result, ultrasonic vibratory rock breaking is more likely to cause damage to the rock when a PDC cutter is used, particularly at a vibration frequency of 40 kHz. As the ultrasonic vibration frequency (20–40kHz) increases, the mechanical specific energy (MSE) initially decreases and then increases. The MSE reaches a minimum value at a frequency of 20–25 kHz, representing a decrease of 15.52%–22.24% compared with conventional rock breaking, which can significantly improve the rock breaking efficiency and reduce the drilling cost. The temperature of the PDC cutter increases significantly under ultrasonic vibration compared with conventional rock breaking. Additionally, the temperature of the PDC cutter increases gradually with an increase in the vibration frequency. These results provide theoretical support for the use of ultrasonic vibration technology.

## Introduction

With increasing human demand for conventional oil and gas resources, shallow resources are nearing exhaustion. Consequently, the exploration and development are gradually shifting from shallow to deep and ultra-deep wells. However, the high strength, abrasiveness, and poor drillability of rocks in deeper formations are gradually becoming more pronounced [1], which dramatically reduces the drilling efficiency of conventional PDC bits. At present, crushing

**Funding:** Shandong Provincial Natural Science Foundation Youth Science Fund of China.

**Competing interests:** The authors have declared that no competing interests exist.

techniques for hard rock can be divided into two types: mechanical energy rock breaking (blasting, drilling, water jet, etc.) [2,3] and thermal energy rock breaking (high-energy pulsed plasma jet, laser, etc.) [4–6]. These methods improve the efficiency of breaking hard rock to a certain extent; however, these drilling methods need to be further optimized because of the limitations of equipment functions and drilling costs. In recent years, ultrasonic vibratory rock breaking technology has attracted considerable attention owing to its advantages of strong penetration, small required cutting force, and fast drilling speed. It is considered a new drilling method that effectively solves the difficult problem of breaking hard rock.

Ultrasonic vibration technology mainly utilizes the phenomenon of resonance to break rocks. When the vibration frequency (above 20 kHz) is close to the intrinsic frequency of rocks (20–40 kHz), resonance will be generated. Resonance causes the vibration inside the rock to reach its peak; as a result, the cracks inside the rock expand rapidly, the compressive strength decreases dramatically, and the rock will be violently broken under the slightest vibration. This can significantly reduce the probability of the stick-slip effect occurring in PDC cutters in hard rock formation to achieve efficient rock breaking [7]. Zhou et al. [8] investigated the fatigue behavior of granite under ultrasonic vibration using a combination of experimental tests and discrete element simulations to explore the crack evolution law of hard rock under ultra-high frequency vibration loading. Zhao et al. [9] used infrared thermography to observe the infrared temperature changes on the surface of granite under ultrasonic vibration, and the results showed that a rapid increase in the infrared temperature signaled that the rock was about to be crushed. The fatigue expansion of the rock under ultrasonic vibration and thermal damage at high temperatures were the main factors determining rock fracture. Wang et al. [10] further verified the feasibility and effectiveness of rock breaking under ultrasonic vibration in practical applications through a series of indoor experiments. The experiments showed that cracks developed rapidly from the inside of the rock under high-speed ultrasonic vibration, resulting in rapid rock damage. Yin et al. [11] tested the compressive strength of rocks by applying various pre-pressures to rock specimens under ultrasonic excitation. They used CT scanning techniques to determine the degree of damage to the rock specimens. The results showed that the degree of damage to the rock specimens under ultrasonic excitation initially increased and then decreased with an increase in pre-pressure. The effects of ultrasonic vibrations on rocks with varying degrees of hardness were studied by adjusting the vibration frequencies. The results showed that the rock-breaking efficiency of the PDC cutter was significantly improved by the introduction of high-frequency axial vibrations. When the vibration frequency was close to the intrinsic frequency of the rock, the cutting force required for rock breaking decreased, resulting in a more stable cutting process. That study also investigated the complex dynamics of ultrasonic impacts on drilling in hard rock [12,13]. Fernando et al. [14,15] conducted experiments on the rotary ultrasonic machining of rocks with three different hardness values. The results showed that the rotary ultrasonic technology could drill at approximately three times the speed of the percussion drilling technology and had a more stable cutting process. Additionally, a model of the mechanical cutting force for rotary ultrasonic machining (RUM) of rocks was developed. This model could be used to predict the relationship between the loading parameters and cutting force. Zhao et al. [16] conducted numerical simulations of granite at varying temperatures under ultrasonic vibration. The results indicated that as the temperature increased, the volume and number of cracks in the rock due to thermal damage also increased. Tang et al. [17] used the discrete element method (DEM) to conduct numerical simulations and investigate the breaking mechanism of rocks under ultrasonic vibrations. Zhang et al. [18] analyzed the individual and combined effects of three loading parameters on the development of granite damage under ultrasonic vibration using analysis of variance.

To date, research on ultrasonic rock breaking has mainly focused on the generation and variation of breaking cracks. However, research on the movement form of the ultrasonic vibration-assisted rock breaking process of the PDC cutter and the dynamic rock breaking efficiency remains relatively scarce, while no research has reported the ultrasonic rock breaking cutting mechanism or the variation trend of the cutting force. This study combines ultrasonic-assisted rock breaking technology with a PDC cutter and uses a three-dimensional finite element model to simulate the rock breaking movement of a PDC cutter under ultrasonic vibration. This study analyzes the variation in the cutting force and temperature of a PDC cutter during the dynamic rock breaking process and discusses the dynamic rock breaking efficiency under different vibration frequencies. Therefore, this study utilizes the Drucker–Prager criterion as the rock yield criterion, establishes a three-dimensional finite element simulation model of rock breaking with a PDC cutter, and employs ABAQUS finite element software to simulate the thermal–structural coupling. This study analyzes and compares rock breaking under ultrasonic vibration and conventional methods using theoretical analyses and numerical simulations. The influence of the vibration frequency on ultrasonic rock breaking is discussed, the rock breaking performance and temperature change of the PDC cutter under various ultrasonic vibration frequencies are analyzed, and the optimal parameter intervals are determined to provide theoretical support for rock breaking with ultrasonic vibration.

## Rock constitutive model and finite element verification

**Constructive modeling and rock failure criterion.** Currently, two codes are commonly used to describe the constitutive relationship during the plastic phase of rock, both domestically and internationally. These codes are the Mohr-Coulomb (M-C) and Drucker-Prager (D-P) codes, respectively. The D-P criterion comprehensively considers the role of hydrostatic pressure and intermediate principal stresses. It can effectively explain the phenomenon of rock yielding under hydrostatic pressure. This study employs the Drucker-Prager (D-P) yield criterion, which can be expressed as follows [19]:

$$\lambda I_1 + \sqrt{J_2} - K = 0 \tag{1}$$

$$I_1 = \sigma_1 + \sigma_2 + \sigma_3 \tag{2}$$

$$J_2 = \frac{1}{6}\left[(\sigma_1 - \sigma_2)^2 + (\sigma_2 - \sigma_3)^2 + (\sigma_3 - \sigma_1)^2\right] \tag{3}$$

$$\lambda = \frac{2\sin\beta}{\sqrt{3}(3 - \sin\beta)} \tag{4}$$

$$K = \frac{6d\cos\beta}{\sqrt{3}(3 - \sin\beta)} \tag{5}$$

where $J_2$ is the stress bias of the second invariant, $I_1$ is the stress force of the first invariant, $\lambda$ and $K$ are experimental constants related to the angle of internal friction $\beta$ of the rock material and the bonding force $d$. $\sigma_1$, $\sigma_2$ and $\sigma_3$ are the first, second, and third principal stresses, respectively. Eq (1) shows that the rock undergoes gradual plastic deformation when in contact with the drill bit. Once the plastic deformation exceeds the critical value, the rock is destroyed and chips begin to dislodge from the rock body. Therefore, the equivalent plastic strain serves as

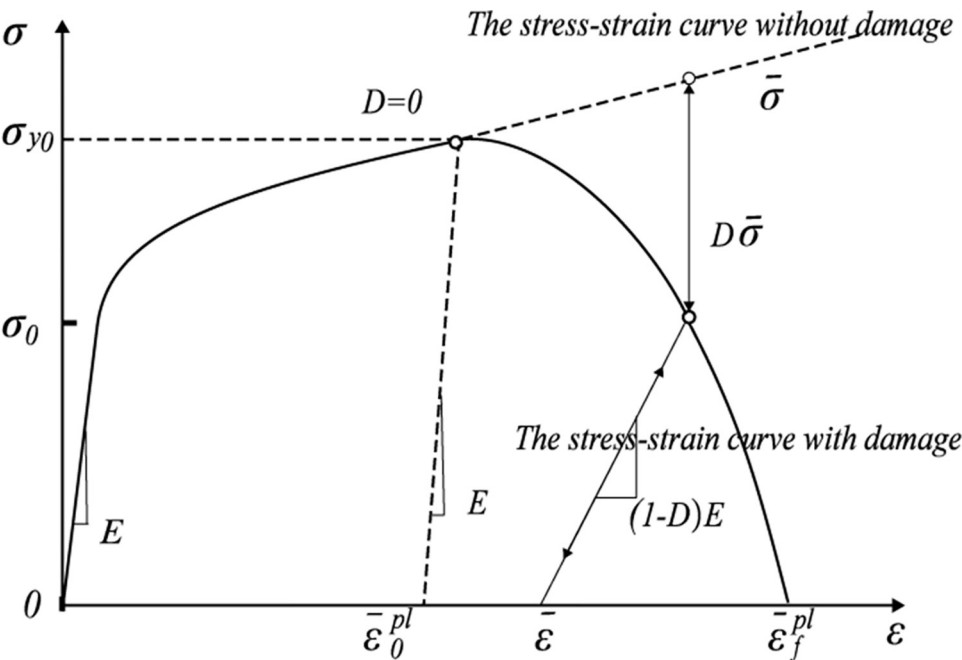

**Fig 1. The stress-strain curve in rock failure process.**

the rock-breaking criterion, that is [20]:

$$\varepsilon^p \leq \bar{\varepsilon}_f^{pl} \tag{6}$$

where $\varepsilon^p$ is the equivalent plastic strain of the rock under external force, $\bar{\varepsilon}_f^{pl}$ is the equivalent plastic strain when the rock chips start to fall off from the rock body.

Fig 1 shows the stress-strain curves during rock failure. The rock-breaking criterion now commonly employs the concept of "Damage Factor D" to describe the damage to the rock and its breaking phenomenon. $\sigma_{y0}$ is the stress at the beginning of the damage, when the total damage produced $D = 0$. When the plastic strain reaches $D = 1$ and the rock is completely breaking [21,22].

**Verification of finite element analysis.** This section presents the use of ABAQUS to simulate uniaxial compression experiments on granite in order to verify the established numerical model. The finite element model comprises a rock sample measuring Φ50mm×100mm and two rigid disks. The lower disk is fixed, and a compressive displacement load of 5 mm is applied to the upper disk in the z direction. The degrees of freedom of the upper disk are constrained in the x and y directions. Embedding cohesive units in rock sample models. Fig 2 shows a comparison of the damage morphology and the compressive stress-axial strain relationship between the rock under numerical simulation and laboratory experimental conditions, which are in close agreement. The simulation results are represented by the red curves, while the blue curves represent the experimental results conducted in the lab. The stress-strain curves obtained from numerical simulation are consistent with the results of laboratory tests. Additionally, the model is capable of simulating rock breaking and stress-strain behavior effectively, as demonstrated by the comparison with the results of uniaxial compression experiments.

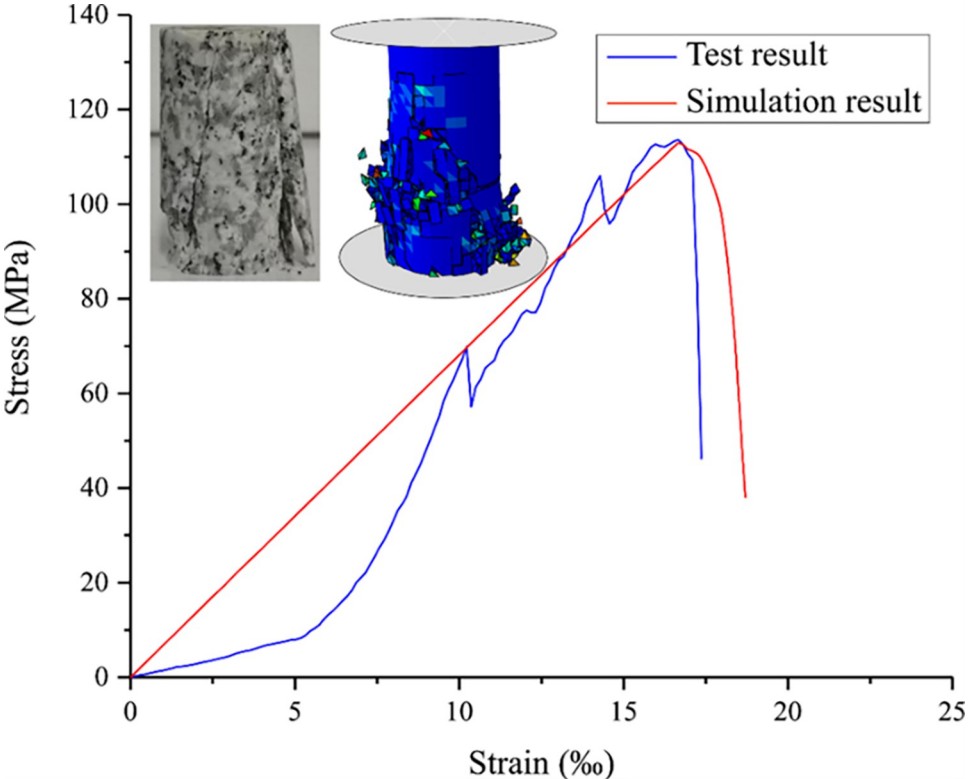

**Fig 2. Comparison of uniaxial compression test and simulation.**

## Numerical model

**Model assumption.** This paper focuses on the temperature and stress changes of PDC cutter during rock breaking under different frequencies of ultrasonic vibration and conventional rock breaking. A simulation model is established to analyze the process. To ensure accurate and efficient simulation results, while also reducing the influence of rock inhomogeneity on experimental results, the following assumptions are conducted:

1. The material of the rock is homogeneous.

2. Ignore the influence of surrounding pressure and drilling fluid column pressure.

3. The cutter is considered a rigid body, and its wear is not considered.

4. Rock units should be removed immediately after breaking, regardless of the effect of rock chips on the breaking process.

**Establishment of PDC cutter and rock modeling.** The PDC cutter is composed of cemented carbide layers (WC-Co) and a polycrystalline diamond layer (PCD). The PDC cutter has a diameter of 13.44 mm. The thickness of the cemented carbide is set to 5 mm, the thickness of the PCD layer is set to 3 mm, and the total thickness of the PDC cutter is set to 8 mm. The back-rake angle of the cutting teeth is set to 15° and the depth of cut is set to 2 mm. In order to avoid the effect of distal constraint on the rock stress, according to the principle of Saint Venant, the rock size is 5–10 times the size of the cutting teeth, and thus the rock size is 100mm×50mm×25mm. Fig 3 shows the finite element model of PDC cutter and rock. To

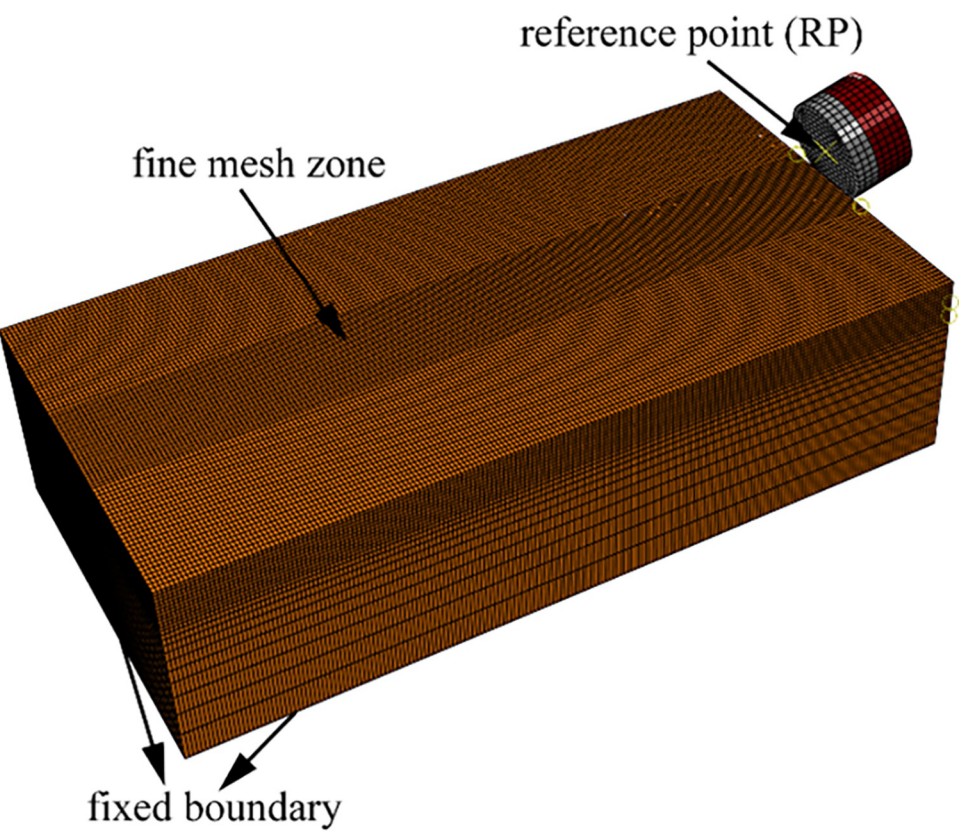

**Fig 3. Finite element modeling of PDC cutter with rock.**

boost the computational accuracy of the measurement, it is necessary to use 8-node hexahedral linear simplified integral element (C3D8RT) to mesh the rock and PDC cutter, and the rock model was divided into 352,000 elements, and the PDC cutter was divided into 1,664 elements. To better simulate the rock-breaking effect of the PDC cutter in practical applications, the local mesh cell size was set to be close to the size of the actual rock particle. The local mesh of the rock-cutting region was refined, and appropriate mesh coarsening was carried out for the part far away from the rock-cutting region. The smallest mesh size was set to 0.5 mm. The physical and mechanical parameters used in the model are listed in Table 1.

**Boundary conditions and constraints.** The simulation uses a full-size drill with a diameter of 76 mm. The drill rotates at 300 rpm. The relationship between linear velocity and rotational speed is as follows:

$$v = \frac{n\pi D}{60} \tag{7}$$

**Table 1. Relevant material parameters used in the finite element analysis.**

| Model material | Density (kg·m⁻³) | Elastic modulus (GPa) | Poisson's ratio | Thermal conductivity W/(m·°C) | Specific heat J/(kg·°C) | Thermal expansion coefficient ($10^{-6}$°C⁻¹) |
|---|---|---|---|---|---|---|
| PCD | 3510 | 897 | 0.07 | 543 | 790 | 2.5 |
| WC-Co | 15000 | 579 | 0.22 | 100 | 230 | 5.2 |
| Rock | 2700 | 45.5 | 0.25 | 2.6 | 881 | 52 |

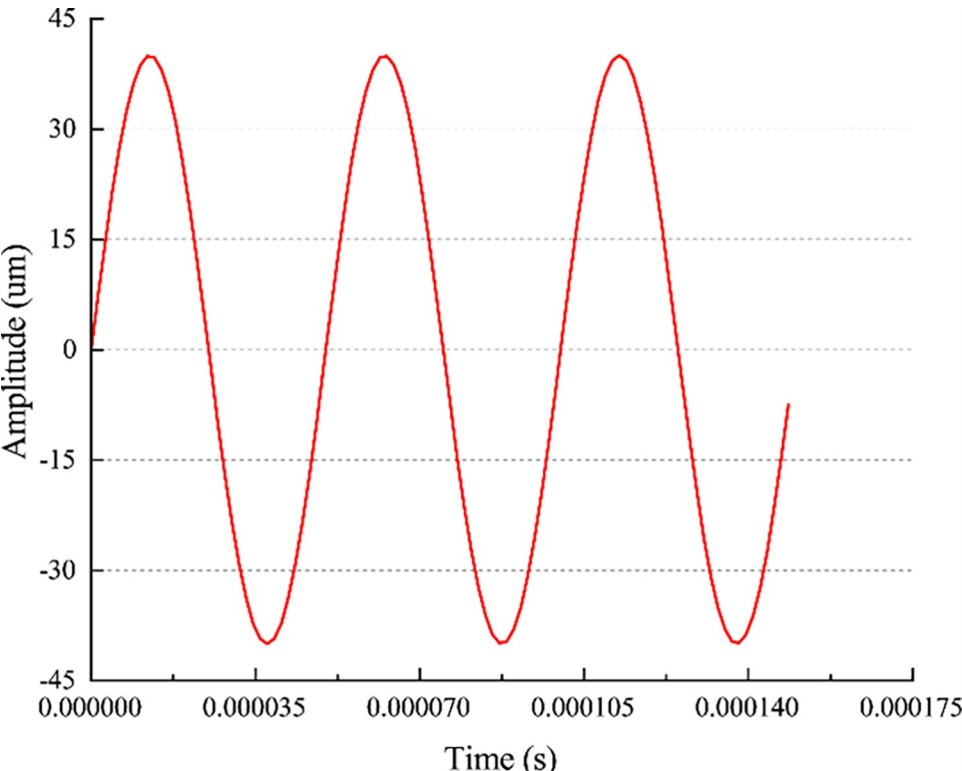

**Fig 4. The amplitude curve.**

where *n* represents the rotational speed of the full-size drill; *D* represents the diameter of the full-size drill; and the cutting speed of the PDC cutter is 1194 mm/s while moving horizontally along the negative direction of the X-axis. The simulation time was 0.08s. Fully secured restraints to the base and perimeter of the rock. Because PDC cutter are much stiffer than rock, the cutter is set as a rigid body and a reference point RP1 is defined. During the simulation process, the PDC cutter and the rock are in face-to-face contact to prevent mesh penetration and errors in the simulation. This allows for a more accurate determination of the contact area and calculation of resulting contact force. The faces with high rigidity are designated as master faces, while those with low rigidity are designated as slave faces. Therefore, the PDC face is designated as the master face and the rock unit set is designated as the slave face. The initial temperature of the simulation was set to 27˚, and the friction coefficient was set to 0.3. The amplitude of ultrasonic vibration is 40um, and the amplitude curve is shown in Fig 4. Since the intrinsic frequency of rocks is 20–40 kHz, we investigated the rock-breaking behavior of PDC cutter at ultrasonic vibration frequencies of 20 kHz, 25 kHz, 30 kHz, 35 kHz, and 40 kHz, respectively.

## Results and discussion

### The process of breaking rocks

In this simulation, the conventional dynamic rock breaking process is compared with the dynamic rock breaking process under ultrasonic vibration. There are two failure modes of rock during the breaking process: ductile failure and brittle failure [23,24]. Rock breaking under the action of ultrasonic vibration occurs mainly in the brittle failure mode [25]. In the

process of rock breaking, the PDC cutter contacts the rock; at this time, the rock undergoes local elastic–plastic deformation without breaking. As a result, the cutting force increases with time, and the small cracks begin to form inside the rock. With an increase in the cutting force, cracks gradually expand along the structural surface of the rock. When the cracks expand to a critical point and the cutting force reaches a critical value exceeding the yield level of the rock, sudden fracture of the entire rock occurs, resulting in rapid fracture of the rock [26]. The rock in front of the PDC cutter undergoes significant shear failure, at which point it is considered a failed unit and removed from the rock body. Consequently, the cutting force decreases rapidly. During each cutting phase, the PDC cutter moves horizontally and experiences a cyclical increase and decrease in cutting force. Therefore, during the cutting process, the cutting force fluctuates continuously over time [27]. As shown in Fig 5, ultrasonic vibration loading does not change the periodic oscillation characteristics of the cutting force. The cutting force reaches its maximum when the rock stress reaches the yield state [28]. Thus, each wave peak corresponds to the maximum cutting force generated by the PDC cutter with the rock in front of it. The average cutting force can be used to assess the ease of rock cutting. The average cutting force of conventional rock breaking is 1055.23 N. The average cutting forces for rock breaking under ultrasonic vibration at frequencies of 20, 25, 30, 35, and 40 kHz are 809.54, 808.26, 843.22, 815.30, and 683.82 N, respectively. These represent decreases of 23.28%, 23.40%, 20.09%, 22.74%, and 35.20%, respectively, compared with conventional rock breaking. The results indicate that the average cutting force generated during rock breaking with ultrasonic vibration decreases significantly, reaching a minimum at a vibration frequency of 40 kHz. Therefore, compared with conventional rock breaking, rock breaking with ultrasonic vibration can significantly improve the stress state of the PDC cutter, which is beneficial for reducing bit wear and extending the service life. Studies have shown that rock has a high compressive strength but lower shear and tensile strengths. The decrease in the average cutting

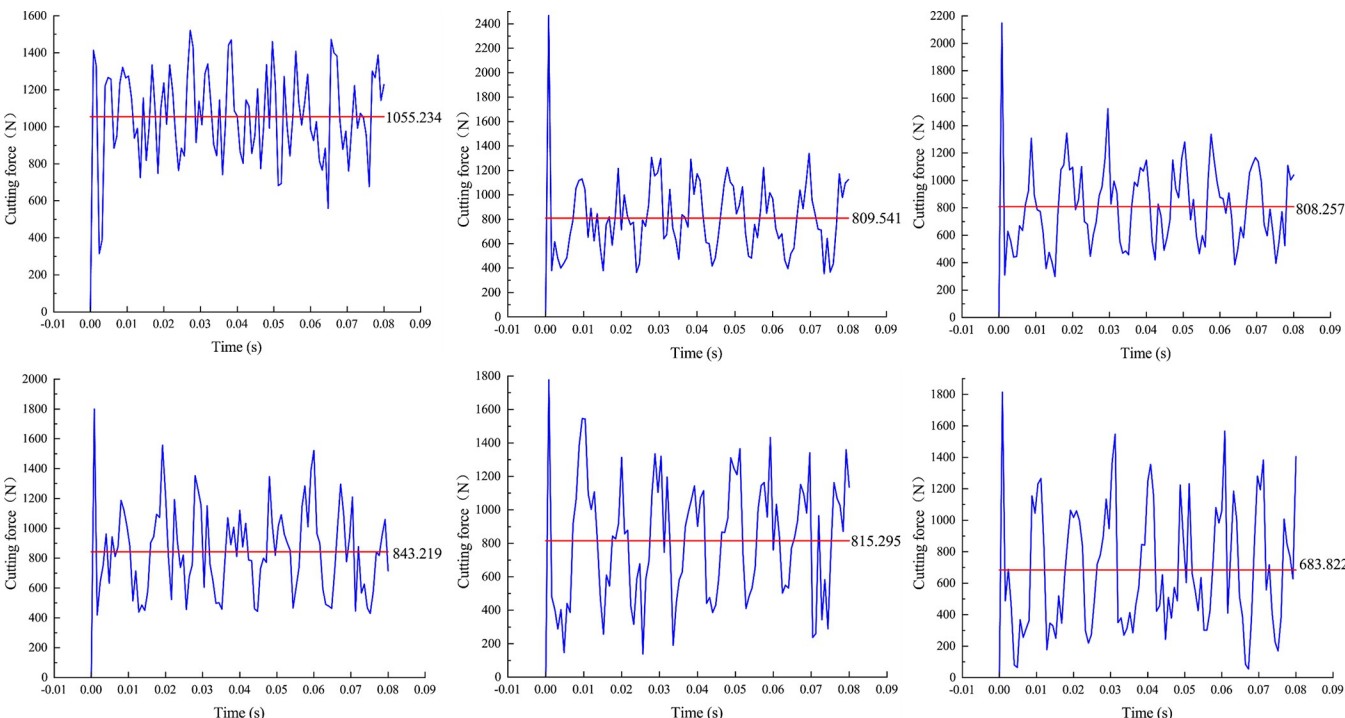

**Fig 5. Cutting force variation under different UHFV loads.** (a) Conventional rock breaking (b) 20kHz (c) 25kHz (d) 30kHz (e) 35kHz (f) 40kHz.

force during ultrasonic vibratory rock breaking is due to the resonance effect of the rock under high-frequency vibration. According to the vibration principle, ultrasonic vibration can produce the effect of compression by pulling on the rock, indirectly producing a tensile force on the rock. When vibrating, the rock will store part of its energy during compression and then produce a tensile force during rebound to achieve the effect of breaking the rock. In the conventional dynamic rock breaking process, the PDC cutter mainly produces shear failure on the rock, whereas in the dynamic rock breaking process with ultrasonic vibration, the PDC cutter mainly produces shear failure on the rock, and tensile failure occurs on the rock through the resonance effect as a supplement. The shear failure is mainly concentrated in the contact area between the PDC cutter and the rock, whereas tensile failure occurs mainly in the lower part of the failure area and tends to extend downward, causing a greater degree of damage to the rock over a wider area.

**Rock stress.** The PDC cutter contacts the rock and produces a localized concentrated force at the rock contact point (red point in Fig 6). The maximum contact stress generated by

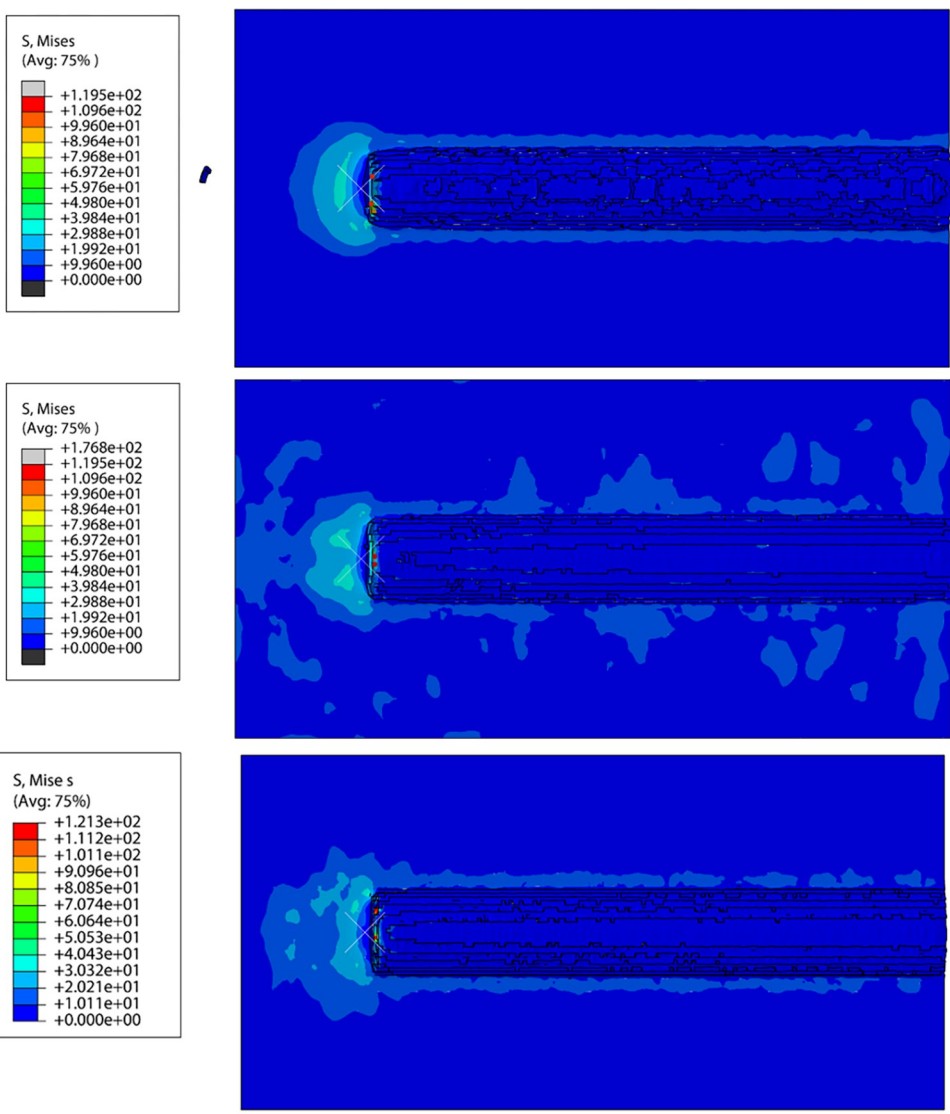

**Fig 6. Comparison of rock stresses in the cutting process.** (a) Conventional rock breaking. (b) 25kHz (c) 40kHz.

the conventional rock breaking process is 119.5 N. However, the rock breaking processes with ultrasonic vibration frequencies of 25 kHz and 40 kHz generate maximum contact stresses of 176.8 N and 121.3 N, respectively. Therefore, in the case of ultrasonic vibration assistance, the resulting stress is clearly higher. When the vibration frequency is 25 kHz, the force is more concentrated, and the stress is increased by 47.9% compared with conventional rock breaking; thus, plastic deformation of the rock can more easily occur. From conventional rock breaking to rock breaking with ultrasonic vibration frequencies of 25 kHz and 40 kHz, the generated stress sequentially increases and then decreases. Thus, it is easiest to break the rock under ultrasonic vibration cutting with a vibration frequency of 25 kHz. This can significantly reduce the degree of wear to the PDC cutter, extend the service life of the PDC cutter, and reduce the drilling cost.

Residual stresses are generated near the edges of the damaged part of the rock. Residual stresses in rock are caused by a uniform material and non-uniform stress action, uniform stress action and non-uniform material, or both. These residual stresses cause plastic deformation of the rock through localized yielding that occurs when the external load is removed. The residual stresses generated in the rock can reduce its strength, causing damage to the unfractured rock and making subsequent fracturing more efficient. During conventional rock breaking, the residual stresses generated in the rock are small, causing less damage to the rock in the vicinity of the PDC cutter and having less impact on the subsequent breaking of the rock. During rock breaking with an ultrasonic vibration frequency of 25 kHz, residual stress is generated in the rock, which can cause damage to a large area of the rock near the PDC cutter. This process significantly improves the rock breaking efficiency in subsequent operations. During the process of breaking rock using 40 kHz ultrasonic vibrations, the generated residual stress range is larger than that of conventional rock breaking, but smaller than that of ultrasonic vibration at a frequency of 25 kHz. The resulting damage to the rock falls between the two methods. The results show that the range of residual stresses generated under ultrasonic vibratory rock breaking at a vibration frequency of 25 kHz is sufficiently large to cause damage to a large area of nearby rock.

## Rock breaking efficiency

MSE is the energy consumed by the PDC cutter in breaking a unit volume of rock, which is an important index to evaluate the performance of the PDC drill bit and rock breaking efficiency; the lower the value of the MSE, the less energy will be consumed in breaking a unit volume of rock, and the higher the rock breaking efficiency will be. Thus, this study utilizes the MSE parameter to analyze the simulated data. The expression is as follows [29]:

$$\text{MSE} = \frac{W}{V} = \frac{F_{\text{h}}d}{Ad} \tag{8}$$

where MSE is the mechanical specific energy (J/m$^3$), $W$ is the total work consumed by the drilling bits, $V$ is the volume of the breaking rock(m$^3$), $F_{\text{h}}$ is the cutting force (N), $A$ is the area of the breaking rock by the PDC cutter (m$^2$); $d$ is the distance traveled by the PDC cutter(m).

In the process of rock breaking, the cutting force fluctuates with time; therefore, the average cutting force and average cutting area are used in this study for convenient calculation, and the rock breaking efficiency is evaluated by comparing the MSE at different vibration frequencies. Fig 7 shows the average cutting force and MSE at different vibration frequencies. The MSE values of the rock breaking with ultrasonic vibration frequencies of 20, 25, 30, 35, and 40 kHz are decreased by 22.24%, 15.52%, 7.09%, 4.24%, and 7.70%, respectively, compared with that of conventional rock breaking. When the vibration frequency is 20 kHz, the MSE has the

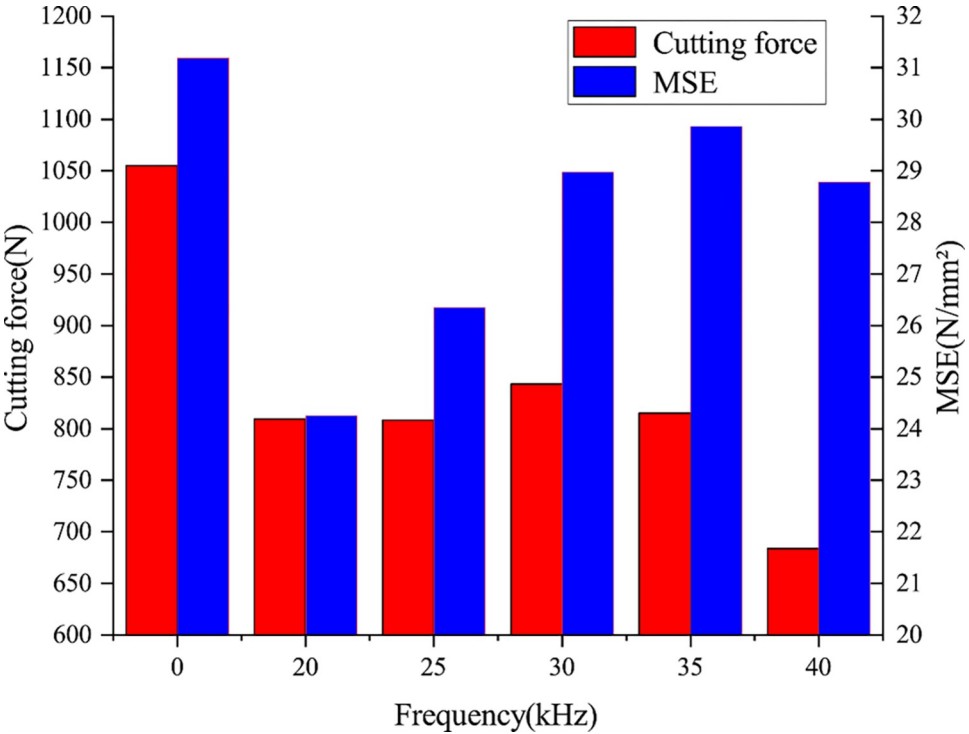

**Fig 7. Average cutting force and MSE at different vibration frequencies.**

minimum value, which can significantly improve the breaking efficiency, reduce energy consumption, and save costs. This is because when the ultrasonic vibration frequency is close to or reaches the intrinsic frequency of the rock (20–40 kHz), resonance occurs, and thus the rock vibration displacement is the largest and is most likely to cause damage. With increasing ultrasonic vibration frequency, the MSE initially decreases and then increases.

When the ultrasonic vibration frequency is lower than the intrinsic frequency of the rock, the response amplitude of the rock is close to that of conventional rock breaking, and the dynamic response of the rock is not sufficiently significant to cause a resonance effect in the rock. When the ultrasonic vibration frequency is higher than the intrinsic frequency of the rock, the rock cannot respond sufficiently rapidly to the high-frequency vibrations owing to the influence of the inertia of the rock medium itself; thus, the response amplitude is very small. When the ultrasonic vibration frequency is close to or reaches the intrinsic frequency of the rock, the dynamic response of the rock is very obvious, and the steady-state response speed, response amplitude, and acceleration of the rock reach the maximum values under the resonance effect. Therefore, rock breaking with ultrasonic vibrations in the appropriate frequency range can significantly improve the rock breaking efficiency.

## Rock breaking temperature

The ultrasonic vibratory rock breaking process in this section uses a vibration frequency of 25 kHz. Fig 8 shows the temperature variation patterns of the PDC cutter under conventional rock breaking and rock breaking with ultrasonic vibration. In the rock breaking process, the thermal conductivity of the PDC layer is more than 20 times that of the rock, and thus friction heat is the main reason for the increase in temperature of the PDC cutter. The PDC cutter rubs against the rock at high speeds under the high-frequency vibration, which causes the

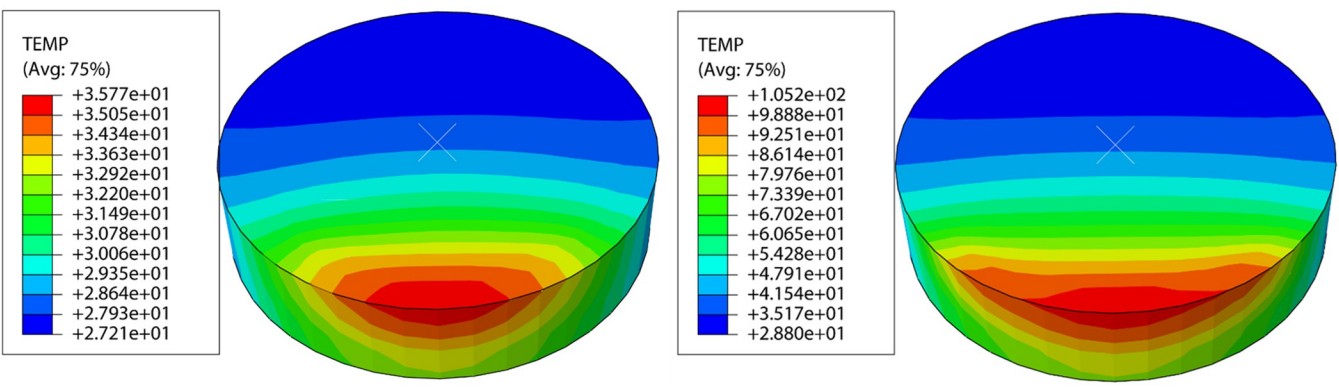

**Fig 8. Temperature nephogram of the PDC cutter.** (a) Conventional PDC cutter (b) PDC cutter at 25kHz vibration.

PDC cutter to be close to the rock. As a result, the heat cannot be rapidly dispersed, and the temperature of the PDC cutter is significantly increased. The warming process of the PDC cutter has three phases: sharp rise, slow rise, and relative stability. As shown in Fig 9, the temperature of the PDC cutter increases rapidly during the initial stage of contact with the rock, after which it increases more slowly during the establishment stage as the PDC cutter stabilizes to break the rock. Finally, the temperature remains relatively stable during the stabilized cutting stage.

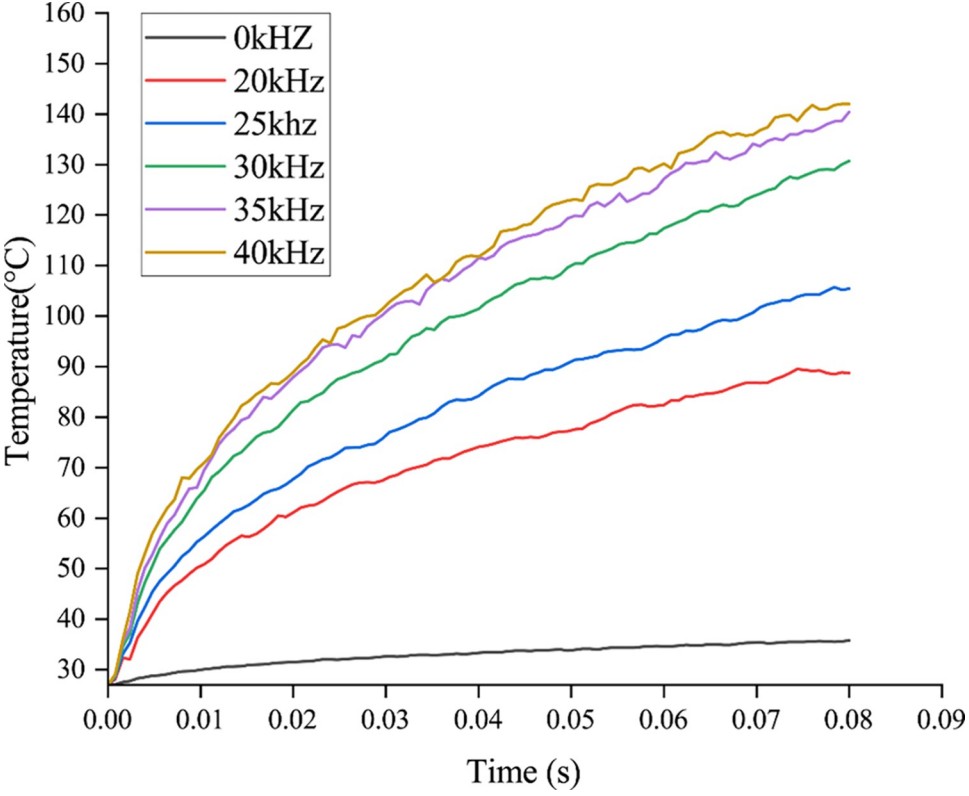

**Fig 9. Temperature variation curves of the PDC cutter under ultrasonic vibration at different frequencies.**

The peak temperatures of the PDC cutter differ between conventional rock breaking and rock breaking with different ultrasonic vibration frequencies. The PDC cutter reaches a peak temperature of 35.77°C during conventional rock breaking. Under rock breaking with ultrasonic vibration frequencies of 20, 25, 30, 35, and 40kHz, the peak temperatures of the PDC cutter are 88.74, 105.41, 130.68, 140.39, and 141.98°C, respectively. The results show that the temperature of the PDC cutter under different frequencies of ultrasonic vibration rock breaking was significantly higher than that of the PDC cutter under conventional rock breaking and that the temperature of the PDC cutter increases with increasing vibration frequency. Under rock breaking with ultrasonic vibration, the friction area of the PDC cutter increases, thereby intensifying the friction between the PDC cutter and the rock. The ultrasonic vibration also increases the relative sliding speed between the PDC cutter and rock, thus increasing the heat generated by friction. Meanwhile, the higher the frequency, the higher the temperature of the PDC cutter will be during rock breaking with ultrasonic vibration. From room temperature to 300°C, the compressive and tensile strengths of the rock increase with increasing temperature [30], indicating that the strength of the rock increases with increasing vibration frequency. To ensure the rock breaking efficiency and improve the service life of the PDC cutter, it is necessary to obtain the optimal vibration frequency for rock breaking. The temperature of the PDC cutter increases by 25.27°C as the vibration frequency increases from 25 kHz to 30 kHz. In addition, the MSE reaches its minimum value between 20 kHz and 25 kHz. Therefore, the results indicate that the optimal frequency for rock breaking is 20–25 kHz under the condition of ensuring a small increase in rock strength.

## Conclusions

To explore the rock breaking mechanism of a PDC cutter under ultrasonic vibration, this study established a three-dimensional dynamic rock breaking model based on finite element software. The dynamic rock breaking process of a PDC cutter under ultrasonic vibration was numerically simulated, and the influence of the vibration frequency on the rock breaking performance of the PDC cutter was investigated and compared with that of the conventional dynamic rock breaking process. The following conclusions were drawn:

(1) Under the action of the ultrasonic vibration load, the average cutting force generated decreased significantly: when the vibration frequency was 40 kHz, the average cutting force decreased by 35.20%, and when the vibration frequency was 20–25 kHz, the average cutting force decreased by 23.28%–23.40%, which can reduce the degree of wear of the PDC cutter and improve its service life of the PDC cutter.

(2) In the dynamic rock breaking process of an ultrasonic vibration-assisted PDC cutter, when the vibration frequency was 25 kHz, the rock was easier to break, and greater residual stress was over a wider area. This can cause fatigue damage to a large are of rock near the PDC cutter and thus significantly improve the rock breaking efficiency.

(3) With increasing frequency of the ultrasonic vibration, the MSE initially decreased and then increased. When the ultrasonic vibration frequency was close to or reached the intrinsic frequency of the rock (20–25 kHz), the MSE decreased by 15.52%–22.24%. Under this condition, the rock resonance effect occurred, and the PDC cutter could break the rock using less energy consumption, which can significantly reduce the cost of drilling.

(4) The temperature of the PDC cutter under ultrasonic vibration assistance was much higher than that of the conventional PDC cutter; as the vibration frequency increased, the temperature of the PDC cutter also increased. The optimal frequency for rock breaking was 20–25 kHz.

The rock breaking mechanism of PDC cutter under the ultrasonic vibration obtained provides theoretical guidance and technical support for ultrasonic vibration technology to solve

hard rock drilling problems in engineering practice, and enables PDC bits to realize a major breakthrough in breaking hard rock.

## Supporting information

**S1 File. The values used to build graphs.**
(DOCX)

## Author Contributions

**Funding acquisition:** Zengzeng Zhang.

**Investigation:** Ruocheng Zhang.

**Methodology:** Zhanfang Huang, Zengzeng Zhang.

**Software:** Ruocheng Zhang, Chunguang Wang.

**Supervision:** Zhanfang Huang, Yalu Han, Zhendong Wang, Chunguang Wang, Qing Yan.

**Writing – original draft:** Ruocheng Zhang.

**Writing – review & editing:** Zengzeng Zhang, Zhendong Wang.

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
