## [Editor Report · Decision Letter 0]

5 Mar 2024

PONE-D-24-07628Research on rock breaking mechanism of PDC cutter under the action of ultrasonic vibrationPLOS ONE

Dear Dr. Zhang,

Thank you for submitting your manuscript to PLOS ONE. After careful consideration, we feel that it has merit but does not fully meet PLOS ONE’s publication criteria as it currently stands. Therefore, we invite you to submit a revised version of the manuscript that addresses the points raised during the review process.

Dear Authors,

I appreciate you considering PLOS ONE journal for your research publication. After thoroughly reviewing the article, I found it very interesting, it seems to be one of the important numerical simulations performed in Abaqus considering rock breaking mechanics. However, I would appreciate it if you could improve its English. I understand that you are not native English speaker, but at the present form, the article English quality is well below the required standard. There are multiple grammar issues. Most of the text follows a compound-complex sentence structure, with syntax errors, punctuation errors, and missing definite articles. At some places I found subject-verb disagreement. Considering these issues, I recommend contacting some English proofreading company and resubmitting this article with improved and error-free English.

We look forward to receiving your revised manuscript.

Kind regards,

Waqas Saleem, Ph.D

Academic Editor

PLOS ONE

Journal Requirements:

   "Shandong Provincial Natural Science Foundation Youth Science Fund of China"

6. We note that your Data Availability Statement is currently as follows: All relevant data are within the manuscript and its Supporting Information files

---

## [Author Response · Author response to Decision Letter 0]

29 Apr 2024

Dear Editors and Reviewers:

Thank you for your letter and for the Reviewers’ comments concerning our manuscript entitled “Research on rock breaking mechanism of PDC cutter under the action of ultrasonic vibration” (Manuscript Number: PONE-D-24-07628). Those comments are all valuable and very helpful for revising and improving our paper, as well as the important guiding significance to our researches. We have studied comments carefully and have made some correction which we hope to meet with approval. Revised portions are marked in red in the paper. The main correction in the paper and the respond to the Reviewer’s comments are as flowing:

Responds to the reviewer ’s comments:

Reviewer #1:

Thank you for reviewing our manuscript so carefully and thanks very much for your positive evaluation which make us confident in our research. At the same time, your precious suggestions will help us to further improve our research. And according to your comment and suggestions, we have revised our content one by one.

Comment 1: This paper does not fully meet PLOS ONE’s publication criteria as it currently stands.

Response: Thank you for your valuable and thoughtful comments, we really agree with your viewpoints. According to your helpful advice, we have made modifications to the format of the paper to meet PLOS ONE’s publication criteria. Thank you again.

Comment 2: English needs to be greatly improved.

Response: Thank you for your valuable advice. We have carefully revised the manuscript according to your comments, and also have re-scrutinized to improve the English by a language polishing service.

Special thanks to you for your good comments and suggestions. 

We tried our best to improve the manuscript and made some changes in the manuscript. These changes will not influence the content and framework of the paper. And here we did not list the changes.

We appreciate for the Editors and Reviewers’ warm work earnestly and hope that the correction will meet with approval. We look forward to hearing from you regarding our submission, and we would be glad to respond to any further questions and comments that you may have.

Once again, thank you very much for your comments and suggestions.

Yours 

Sincerely

2024.04.29

---

## [Editor Report · Decision Letter 1]

9 Jun 2024

PONE-D-24-07628R1Research on rock breaking mechanism of PDC cutter under the action of ultrasonic vibrationPLOS ONE

Dear Dr. Zhang,

Thank you for submitting your manuscript to PLOS ONE. After careful consideration, we feel that it has merit but does not fully meet PLOS ONE’s publication criteria as it currently stands. Therefore, we invite you to submit a revised version of the manuscript that addresses the points raised during the review process.

**Editor's Note/Recommendation: **I believe the authors are not native speakers, but this manuscript has serious issues regarding sentence structure, clarity, coherence, syntax errors, punctuation, misuse of adjectives, and appropriate tense. The manuscript must be sent to an authentic English editing service before submitting the revised version.

We look forward to receiving your revised manuscript.

Kind regards,

Waqas Saleem, Ph.D

Academic Editor

PLOS ONE

---

## [Author Response · Author response to Decision Letter 1]

23 Jul 2024

Dear Editors and Reviewers:

Thank you for your letter and for the Reviewers’ comments concerning our manuscript entitled “Research on rock breaking mechanism of PDC cutter under the action of ultrasonic vibration” (Manuscript Number: PONE-D-24-07628). Those comments are all valuable and very helpful for revising and improving our paper, as well as the important guiding significance to our researches. We have studied comments carefully and have made some correction which we hope to meet with approval. Revised portions are marked in red in the paper. The main correction in the paper and the respond to the Reviewer’s comments are as flowing:

Responds to the reviewer ’s comments:

Thank you for reviewing our manuscript so carefully and thanks very much for your positive evaluation which make us confident in our research. At the same time, your precious suggestions will help us to further improve our research. And according to your comment and suggestions, we have revised our content one by one.

Comment 1: Line 13-15; the word “great” has used in abstract. Use some other suitable word as engineering/ technical point of view.

Response: Thank you for your valuable and thoughtful comments, we really agree with your viewpoints. We have corrected the sentence according to your comment in the revised manuscript. Thank you again.

Modified sentence

This is important for practical engineering applications.

Comment 2: Line 14; there is mention in abstract as “significantly reduce the cost”. How it is happened and needs clarification as you have not mention any data to support in this context?

Response: Thank you for your valuable comments. During the breaking process, when the ultrasonic vibration reaches the optimal frequency, the required cutting force is minimized and the energy consumption is reduced, which significantly reduces the cost of rock breaking and extends the service life of the PDC cutter.

Comment 3: Use mechanical specific energy (MSE) in Line 22 instead of Line 23 of abstract.

Response: We are sorry for this mistake. We have corrected the sentence according to your comment.

Modified sentence

As ultrasonic vibration frequency (20~40kHz) increases, the mechanical specific energy (MSE) initially decreases and then increases. At a frequency of 20~25kHz, the MSE reaches its minimum value.

Comment 4: Line 30; Use “ultrasonic frequency or ultrasonic vibration frequency” instead of “ultrasonic”. 

Response: Thank you for pointing out the problem. According to your comment, we have corrected the “ultrasonic” into “ultrasonic frequency” in the revised manuscript.

Modified sentence

Keywords: PDC cutter; ultrasonic frequency; temperature; rock breaking efficiency

Comment 5: Line 32-34; “and” has used 4 times in one sentence. Improve this sentence and break into small meaningful sentences.

Response: Thank you for your valuable advice. We have carefully revised the manuscript according to your comments, and also have re-scrutinized to improve the English by a language polishing service.

Modified sentence

With increasing human demand for conventional oil and gas resources, shallow resources are nearing exhaustion. Consequently, the exploration and development are gradually shifting from shallow to deep and ultra-deep wells.

Comment 6: Line 72; rotary ultrasonic technology in drilling seems not have relation with rock-breaking with PDC cutter. Please elaborate or improve the words in manuscript.

Response: Thank you for your valuable comments. Polycrystalline diamond composite (PDC) bits are the main rock-breaking tools for exploration nowadays, which are widely used in the exploration and development process because of their high rock-breaking efficiency and fast drilling speed. However, the breaking effect of PDC bits is not good in deep hard rock drilling. Therefore, this paper combines ultrasonic vibratory rock-breaking technology with PDC bits to adapt to the challenges posed by deeper formations, so that it can realize cutting and crushing while high-frequency fatigue rock-breaking occurs in deep hard-rock formations, which is of significance for improving the breaking efficiency of hard rock, saving drilling costs and solving the difficult problem of hard-rock drilling.

Comment 7: Line 84-88; it is required to elaborate in small sentences for making it more understandable / clarity. 

Response: Thank you for pointing out the problem, we have made changes in the manuscript.

Modified sentence

To date, research on ultrasonic rock breaking has mainly focused on the generation and variation of breaking cracks. However, research on the movement form of the ultrasonic vibration-assisted rock breaking process of the PDC cutter and the dynamic rock breaking efficiency remains relatively scarce, while no research has reported the ultrasonic rock breaking cutting mechanism or the variation trend of the cutting force.

Comment 8: Line 108 and Line 120; there is need to add reference as previous research equations have applied.

Response: We sincerely appreciate the valuable comments. As suggested by your, we have checked the literature carefully and added references in the revised manuscript.

This study employs the Drucker-Prager (D-P) yield criterion, which can be expressed as follows [19]:

19. Huang J, Zeng B, He Y, Wang X, Qian L, Xia C, et al. Numerical study of rock-breaking mechanism in hard rock with full PDC bit model in compound impact drilling. Energy Reports. 2023;9:3896-909. doi: 10.1016/j.egyr.2023.02.084.

Therefore, the equivalent plastic strain serves as the rock-breaking criterion, that is [20]:

20. Liu W, Zhu X, Li B. The rock breaking mechanism analysis of rotary percussive cutting by single PDC cutter. Arabian Journal of Geosciences. 2018;11(9). doi: 10.1007/s12517-018-3530-6.

Comment 9: Line 175; the caption of Fig 3 is bold. It is required to improve

Response: Thank you for pointing out the problem. Due to our negligence, the font of the caption was confused. We have carried out all checks and modifications in the manuscript.

Comment 10: Line 193; “In this simulation, we only need to investigate”. There is need to improve words.

Response: Thank you for your valuable comments. We have re-written this part according to your suggestion.

Modified sentence

Since the intrinsic frequency of rocks is 20–40 kHz, we investigated the rock-breaking behavior of PDC cutter at ultrasonic vibration frequencies of 20 kHz, 25 kHz, 30 kHz, 35 kHz, and 40 kHz, respectively.

Comment 11: Line 287; there is need give reference of Equation quoted from literature.

Response: Thank you for your valuable comments. According to your suggestion, reference has been added.

The expression is as follows[29]:

29. Liu S, Zhou F, Li H, Chen Y, Wang F, Guo C. Experimental Investigation of Hard Rock Breaking Using a Conical Pick Assisted by Abrasive Water Jet. Rock Mechanics and Rock Engineering. 2020;53(9):4221-30. doi: 10.1007/s00603-020-02168-2.

Comment 12: Fig 3; text is too small and not readable. It is required to improve Fig.3. Also, improve the text of all the figures as per Journal formate.

Response: Thanks for your valuable comments, we have checked the formatting of all the figures in the manuscript and made changes in the new manuscript.

Fig 3. Finite element modeling of PDC cutter with rock

Comment 13: In rock breaking, vibration technology is using since long time. What is the novelty of this research work?

Response: Thank you for your valuable comments. Currently, most of the research on vibration technology is mainly focused on the theory and macro-experimentation. There has been no relevant discussion among researchers on the cutting mechanism of ultrasonic rock fragmentation and the variation law of cutting force. The innovation point of this paper is to combine the ultrasonic vibration technology with the widely used PDC bits for drilling, and study the changing law of cutting force and MSE of the cutter in the process of ultrasonic vibration rock breaking, in order to improve rock breaking efficiency.

Comment 14: What is accuracy of the simulated model as compared to actual experimental results?

Response: Thank you for your valuable comments. ABAQUS was used to simulate the uniaxial compression of the granite model. The parameters of the model were calibrated through uniaxial compression experiments. The physical and mechanical parameters used in the model are listed in Table 1. The finite element analysis used a Ф50 mm × 100 mm rock sample and two rigid disks. A completely fixed constraint was applied to the bottom plate, and a compressive displacement load of 5 mm was applied to the upper plate, which only had a z-directional degree-of-freedom. Fig. 7 shows the failure pattern of the rock under uniaxial compression. Fig. 8 shows the stress–strain curves obtained from the numerical simulations and laboratory experiments. These results show that the simulated stress–strain curve is consistent with the results of laboratory tests. The peak strengths of the two curves and the slope of the curve at the elastic stage are basically consistent, and both curves showed the brittle failure characteristics after the peak. A comparison of the numerical simulation and experimental results of uniaxial compression demonstrate that this model can adequately simulate the fracturing and stress–strain behavior of the rock.

Comment 15: Before this research, is there any research work is available in literature which compare conventional and ultrasonic rock breaking?

Response: Thank you for your valuable comments. Wiercigroch and Fernando et al. have conducted similar experimental studies.

Wiercigroch et al. conducted experimental tests on the efficiency of conventional rock drilling methods and ultrasonic impact. The test results showed that the drilling speed was significantly increased after applying ultrasonic vibratory loads compared to the conventional rotary drilling method. Fernando et al. used a rotary ultrasonic machining unit (RUM) for rock crushing tests, and the mechanical drilling speed was about three times that of percussive drilling.

Wiercigroch M., J. Wojewoda, and A.M. Krivtsov. Dynamics of ultrasonic percussive drilling of hard rocks. Journal of Sound and Vibration, 2005. 280(3 5): 739 757.

Fernando P. Zhang M., and Pei Z.J. Rotary ultrasonic machining of rocks: An experimental investigation. Advances in Mechanical Engineering, 2018. 10(3).

Fernando P., Pei Z.J., and Zhang M. Mechanistic cutting force model for rotary ultrasonic machining of rocks. The International Journal of Advanced Manufacturing Technology, 2020. 109(12): 109 128.

Comment 16: In results, there is need to compare similar research works which already reported in literature. 

Response: Thank you for your valuable comments. Necessary change in the statements has been made in the revised manuscript.

Comment 17: How this research work has saved the cost? However, there is not presented any data related to the cost?

Response: Thank you for your valuable comments. The cost savings are reflected by the cutting force of the PDC cutter and mechanical specific energy (MSE).

(1) The average cutting force of PDC cutter is 1055.23 N. After applying different frequencies of ultrasonic vibration, the average cutting force of PDC cutter is 809.54 N, 808.26 N, 843.22 N, 815.30 N, and 683.82 N, respectively. These represent decreases of 23.28%, 23.40%, 20.09%, 22.74%, and 35.20%, respectively, compared with conventional rock breaking. Obviously, under ultrasonic vibration rock breaking, the average cutting force decreases significantly, which is conducive to reducing the wear of the drill bits and prolonging its service life. This is the embodiment of material savings.

(2) The MSE of rock breaking under ultrasonic vibration rock breaking frequency of 20kHz, 25kHz, 30kHz, 35kHz, 40kHz is decreased by 22.24%, 15.52%, 7.09%, 4.24%, 7.70%, respectively, compared with that of conventional rock breaking. This is a reflection of reduced energy consumption.

Comment 18: What is the optimal ultrasonic vibration frequency by considering cutting force, breaking efficiency, temperature? On which frequency value the process is required to be carried out?

Response: Thank you for your valuable comments. The temperature of the PDC cutter increases by 25.27 °C as the vibration frequency increases from 25 kHz to 30 kHz. In addition, the MSE reaches its minimum value between 20 kHz and 25 kHz. Therefore, the results indicate that the optimal frequency for rock breaking is 20–25 kHz under the condition of ensuring a small increase in rock strength. 

Comment 19: In conclusions section the last paragraph (2-3 lines) is required to describe practical application of this research work. 

Response: Thank you for your valuable comments. Also thank you for pointing out the problem, which is of great help to the integrity of our paper, we have made changes in the manuscript.

The rock breaking mechanism of PDC cutter under the ultrasonic vibration obtained provides theoretical guidance and technical support for ultrasonic vibration technology to solve hard rock drilling problems in engineering practice, and enables PDC bits to realize a major breakthrough in breaking hard rock.

General

Comment 20: The word “the” has used much where also not needed. Please check and remove where not required. 

Response: Thank you for your valuable comments. Necessary change in the statements has been made in the revised manuscript.

Comment 21: Long sentences are not understandable. There is need to make sentences as per English language and grammar.

Response: Thank you for your valuable comments, we have checked all the grammar

in the manuscript, and also have improved the English writing by Elsevier's language editing service in the revised manuscript.

Comment 22: In manuscript, the abbreviations are required to write as per format of journal.

Response: Thank you for your comments on our paper. We have corrected it in the revised manuscript.

For example, polycrystalline diamond compact (PDC), mechanical specific energy (MSE)

Comment 23: English language is required to be improved of complete manuscript.

Response: Thank you for your valuable advice. We have carefully revised the manuscript according to your comments, and also have improved the English writing by Elsevier's language editing service in the revised manuscript.

Comment 24: Formatting of the manuscript is required as per journal requirements.

Response: Thank you for your valuable comments, we have revised the formatting in the new manuscript according to the journal requirements.

Comment 25: References are required to be format as per journal requirements.

Response: Thank you for your valuable comments, we have revised the references in the new manuscript according to the journal requirements.

Special thanks to you for your good comments and suggestions.

We tried our best to improve the manuscript and made some changes in the manuscript. These changes will not influence the content and framework of the paper. And here we did not list the changes.

We appreciate for the Editors and Reviewers’ warm work earnestly and hope that the correction will meet with approval. We look forward to hearing from you regarding our submission, and we would be glad to respond to any further questions and comments that you may have.

Once again, thank you very much for your comments and suggestions.

Yours 

Sincerely

2024.07.24

---

## [Editor Report · Decision Letter 2]

25 Jul 2024

Research on rock-breaking mechanism of PDC cutter under the action of ultrasonic vibration

PONE-D-24-07628R2

Dear Dr. Zhang,

We’re pleased to inform you that your manuscript has been judged scientifically suitable for publication and will be formally accepted for publication once it meets all outstanding technical requirements.

An invoice will be generated when your article is formally accepted. Please note that if your institution has a publishing partnership with PLOS and your article meets the relevant criteria, all or part of your publication costs will be covered. Please make sure your user information is up-to-date by logging into Editorial Manager at Editorial Manager® and clicking the ‘Update My Information' link at the top of the page. If you have any questions relating to publication charges, please contact our Author Billing department directly at authorbilling@plos.org.

If your institution or institutions have a press office, please notify them about your upcoming paper to help maximize its impact. If they’ll be preparing press materials, please inform our press team as soon as possible—no later than 48 hours after receiving the formal acceptance. Your manuscript will remain under strict press embargo until 2 pm Eastern Time on the date of publication. For more information, please contact onepress@plos.org.

Kind regards,

Waqas Saleem, Ph.D.

Academic Editor

PLOS ONE

---

## [Editor Report · Acceptance letter]

8 Dec 2024

PONE-D-24-07628R2 

PLOS ONE

Dear Dr. Zhang, 

I'm pleased to inform you that your manuscript has been deemed suitable for publication in PLOS ONE. Congratulations! Your manuscript is now being handed over to our production team.

Kind regards, 

on behalf of

Dr. Waqas Saleem 

Academic Editor

PLOS ONE